# Narrative Order Aware Story Generation via Bidirectional Pretraining Model with Optimal Transport Reward

**Zhicong Lu**[1,2], **Li Jin**[1,†] **Guangluan Xu**[1], **Linmei Hu**[3], **Nayu Liu**[4]
**Xiaoyu Li**[1], **Xian Sun**[1], **Zequn Zhang**[1], **Kaiwen Wei**[1,2]

[1] Key Laboratory of Network Information System Technology (NIST), Aerospace
Information Research Institute, Chinese Academy of Sciences
[2] School of Electronic, Electrical and Communication Engineering, University of Chinese Academy of Sciences
[3] Beijing Institute of Technology   [4] School of Computer Science and Technology, Tiangong University
{nazaritelzc, jinlimails}@gmails.com

## Abstract

To create a captivating story, a writer often plans a sequence of logically coherent events and ingeniously manipulates the narrative order to generate flashback in place. However, existing storytelling systems suffer from both insufficient understanding of event correlations and inadequate awareness of event temporal order (e.g., go to hospital *<after>* get ill), making it challenging to generate high-quality events that balance the logic and narrative order of story. In this paper, we propose a narrative order aware framework BPOT (**B**idirectional **P**retraining Model with **O**ptimal **T**ransport Reward) for story generation, which presents a bidirectional pretrained model to encode event correlations and pairwise event order. We also design a reinforcement learning algorithm with novel optimal transport reward to further improve the quality of generated events in the fine-tuning stage. Specifically, a narrative order aware event sequence model is pretrained with the joint learning objectives of event blank infilling and pairwise order prediction. Then, reinforcement learning with novel optimal transport reward is designed to further improve the generated event quality in the fine-tuning stage. The novel optimal transport reward captures the mappings between the generated events and the sentences in the story, effectively measuring the quality of generated events. Both automatic and manual evaluation results demonstrate the superiority of our framework in generating logically coherent stories with flashbacks.

## 1 Introduction

As the fundamental element of numerous literary genres, stories have played a significant role in various social domains, such as literature, education, and entertainment. However, manually creating high-quality stories is often time-consuming and laborious. Therefore, developing an AI system that

---

[†]Corresponding author.

| Input | the police were trying to catch a neighborhood thief. |
|---|---|
| Fine-tuned Bart (Lewis et al., 2020) | the thief was wearing a mask. the mask was fake. the police caught the thief. he was arrested. |
| Megatron-124M (Xu et al., 2020) | they had not arrested him yet. they put him in a jail cell for a month. the jail cell was full of people. they brought him to a jail cell. |
| Flashback (Han et al., 2022) | *the thief had stolen a wallet from a neighbor's car.* the police arrested the thief. he was sentenced to community service. now, the police are trying to apprehend the thief! |

Table 1: Stories generated by existing storytelling systems. The red texts and brown texts indicate logical incoherence and flashback respectively.

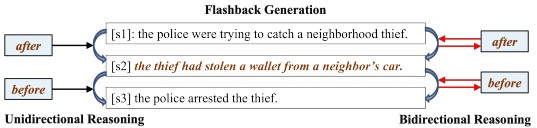

Figure 1: An example of flashback generation. Temporal prompts (*<before>*, *<after>*) hint at the pairwise event order, thereby manipulating the narrative order to generate flashbacks (the brown texts) in place.

can generate stories like human writer is meaningful. Story generation aims to generate a series of events and organize them into a reasonable story given limited information (Huang et al., 2021).

Recently, a significant process has been made in story generation. Similar to human writers, Martin et al. (2018); Yao et al. (2019); Xu et al. (2018) introduced a planning-based method, namely planning sketch prior to writing story. To improve the controllability of story generation (Kong et al., 2021; Xie et al., 2022), different attributes (e.g., style, psychology) were added to sketch. Besides, Guan et al. (2020); Xu et al. (2020) leveraged external knowledge bases to generate commonsense stories. Although existing methods can generate fluent stories related to the desired attribute, they pay little attention to event correlations, resulting in insufficient understanding of event correlations. This makes it difficult to maintain logical coherence as a whole story (Zhou et al., 2022c). Moreover, they rarely consider the crucial role of narrative order in story generation. For example in Table 1,

all storytelling systems exhibit logical incoherence and only the last system reverses narrative order in place to generate flashback.

Flashback is a common writing technique that describes the *past* events to provide context for the current narrative. It arouses reader's interest and deepens their understanding of the narratives. To generate flashback in story, Han et al. (2022) proposed temporal prompt. As shown in Figure 1, *<after>* hints at the pairwise event order, which triggers model to reverse the narrative order to describe the *past* events for flashback generation ("catch a neighborhood thief" occurs later than "had stolen a wallet"). However, they face the problems of inefficient prompt and deteriorating logic in some cases (shown in section 4.5). It is probably because model always unidirectionally reasons event from order, lacking the process of inversely inferring order from pairwise events, resulting in inadequate awareness of event temporal order. Additionally, when generating high-quality events in story, it is intrinsically more challenging to balance the logic and narrative order than merely focusing on logic.

In this paper, we propose a narrative order aware framework BPOT for story generation, which presents a bidirectional pretrained model to encode event correlations and pairwise event order. We also design a reinforcement learning (RL) algorithm with novel optimal transport (OT) reward to further improve the quality of generated events in the fine-tuning stage. Specifically, a narrative order aware event sequence model is pretrained with the joint learning objectives of event blank infilling and pairwise order prediction. Then, we employ OT to capture the mappings between the generated events and the sentences in the story. Based on the mappings, we construct a novel reward to effectively measure the quality of generated events, further improving the generated event quality under the optimization of the RL algorithm. In summary, our main contributions include:

1) We propose a narrative order aware framework BPOT for story generation, which pretrains an event sequence model with the joint learning objectives of event blank infilling and pairwise order prediction to encode the event correlations and pairwise event order.

2) We design an RL algorithm with novel OT reward to further improve the quality of generated event in the fine-tuning stage. The novel OT reward effectively measures the generated event quality

based on the mappings captured by OT.

3) Extensive evaluation results demonstrate the superiority of our framework in generating logically coherent stories with flashbacks.

## 2 Preliminaries

This section provides a description of the sketch used in this work, followed by an introduction to the basic concept of OT.

### 2.1 Sketch Details

We follow the line of the planning-based method (Martin et al., 2018; Yao et al., 2019) for story generation. Our sketch $S$ is composed of alternating splicing of event $e$ and temporal prompt $t$, where $t \in \{<before>, <after>, <vague>\}$. It hints at the pairwise event order, thereby manipulating the narrative order. Note that *<vague>* represents arbitrary event order. Formally, an event $e$ is a structure $(a^0, v, a^1)$, which is extracted from the corresponding sentence $s$ in story $Y$ by semantic role labeling (SRL) tools (Gardner et al., 2018). $v$ is the trigger (Wei et al., 2021) describing the event, $a^0$ and $a^1$ are its relevant arguments (Wei et al., 2022). We only consider one event in each sentence for simplicity. Let $e_{i,n} = (a^0_{i,n}, v_{i,n}, a^1_{i,n})$ denotes the $n$-th event in $i$-th sketch $S_i$, $t_{i,n}$ denotes the event order between the $n$-th event and the $(n+1)$-th event in $S_i$. Then a sketch with $m$ events can be represented as $S_i = \{e_{i,1}, t_{i,1}, ..., e_{i,m}, <eoe>\}$, where *<eoe>* refers to event ending. A sketch with five events is shown at the bottom of Figure 3.

### 2.2 Optimal Transport

OT has recently been introduced to numerous tasks in NLP (Wei et al., 2023). Its original objective is to find the OT maps between two data distributions with minimum cost (Kantorovich, 2006). Formally, given two complete metric spaces X and Y, let $u(x)$ and $v(y)$ denote two discrete probability distributions on X and Y respectively, where $\sum_{i=1}^{n} u(x_i) = \sum_{j=1}^{m} v(y_j) = 1$. Then the OT maps $T^*$ between $u(x)$ and $v(y)$ is obtained by solving the optimization problem (1):

$$T^* = \arg\min_{T} \sum_{i,j=1}^{} t_{ij} \cdot c(x_i, y_j)$$
$$s.t. \sum_{j=1}^{} t_{ij} = u(x_i) \quad \forall i \in \{1, ..., n\}$$
$$\sum_{i=1}^{} t_{ij} = v(y_j) \quad \forall j \in \{1, ..., m\} \quad (1)$$
$$T \in R_+^{n \times m}$$

where $c$ is the cost function. To effectively solve (1), researchers have proposed some approximation algorithms, such as Sinkhorn (Cuturi, 2013), IPOT (Xie et al., 2020). In this work, we adopt IPOT.

## 3 Methodology

In this section, we detail our proposed narrative order aware framework BPOT for story generation, which presents a bidirectional pretrained model to encode event correlations and pairwise event order. As shown in the upper of Figure 3, an RL algorithm with novel OT reward is designed to further improve the generated event quality. Particularly, a narrative order aware event sequence model is pretrained with the joint learning objectives of event blank infilling and pairwise order prediction as shown in Figure 2. In the fine-tuning stage, we employ OT to capture the mappings between the generated events and the sentences in the story. Based on the mappings, a novel reward is constructed to effectively measure the quality of generated events, further improving the generated event quality under the RL optimization.

### 3.1 Vanilla Pipeline Generation

The vanilla pipeline generation of our framework is shown at the bottom of Figure 3. Following the line of the planning-based method, we plan sketch prior to writing story. Firstly, golden sketch $\hat{S}_i$ is extracted from golden story $\hat{Y}_i$ as mentioned in section 2.1. Temporal prompts are obtained by identifying the pairwise event order between sentences by ECONET (Han et al., 2020) in advance. Then all events are masked in $\hat{S}_i$ except for the input event to obtain input $x_i^t$. Last, we require sketch model to generate sketch $S_i$ given on $x_i^t$ and story model to unfold $S_i$ into story $\hat{Y}_i$. The pretrained BART-base (Lewis et al., 2020) is served as both sketch model and story model. Formally, Let $\alpha$ and $\theta$ denote the parameters of sketch model and story model respectively, per sample loss for two models during training can be expressed as (2), (3):

$$L_\alpha = -logp(\hat{S}_i|x_i^t) = -\sum_{k=1}^{|\hat{S}_i|} logp(\hat{S}_{i,k}|x_i^t, \hat{S}_{i,<k}) \quad (2)$$

$$L_\theta = -logp(\hat{Y}_i|S_i) = -\sum_{k=1}^{|\hat{Y}_i|} logp(\hat{Y}_{i,k}|S_i, \hat{Y}_{i,<k}) \quad (3)$$

We use $S_i$ rather than $\hat{S}_i$ to reduce the discrepancy between training and inference.

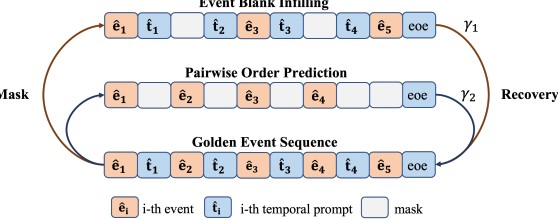

Figure 2: An illustration of the bidirectional pretraining of event blank infilling and pairwise order prediction

### 3.2 Bidirectional Pretraining

Intuitively, the quality of sketch greatly influences the generated story. However, due to suffering from both insufficient understanding of event correlations and inadequate awareness of event temporal order, it is challenging for sketch model to generate high-quality events that balance the logic and narrative order of story. Therefore, we present a joint pretraining of event blank infilling and pairwise order prediction to encode event correlations and pairwise event order as shown in Figure 2.

**Event Blank Infilling** demands model to reason blank event from the given pairwise event order. It requires the generated event to not only be reasonable in event correlations but also conform to a specific event order. Concretely, for each event $\hat{e}_i$ in $\hat{S}_i$, we mask it with a probability of 0.25 to obtain $S_i^e$ and keep all temporal prompts $\hat{t}$ visible. If no events are selected for masking, we randomly choose one to mask. By doing so, the model is able to pay more attention to understanding event correlations conditioned on a specific order.

**Pairwise Order Prediction** requires model to reason order from the given pairwise events within the generated paradigm. Concretely, we mask all temporal prompts $\hat{t}$ in $\hat{S}_i$, where *<eoe>* is also masked to facilitate model to understand the story ending. To prevent excessive deviation from the sketch generation, we additionally mask an event to obtain $S_i^t$. If an event is chosen to mask, the corresponding temporal prompt or *<eoe>* will be restored. Since only one event is masked, model reasons pairwise event order in most cases, which facilitates model to have a better awareness of event temporal order.

To jointly encode event correlations and event temporal order, we first obtain $S_i^e$ and $S_i^t$ from $\hat{S}_i$ for each sample. Then, we execute different learning objectives in parallel and combine their loss in a varying ratio to jointly guide optimization. Per sample loss $L_p$ can be represented as (4):

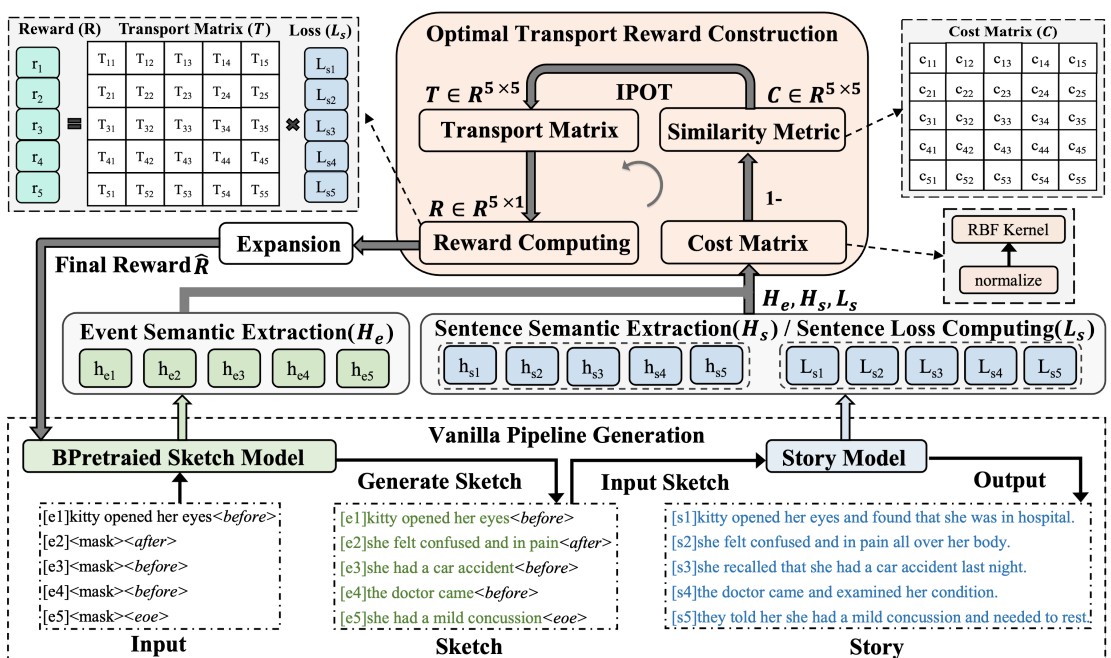

Figure 3: An illustration of our framework for story generation. We take an example of a five-sentence story. The bottom shows the vanilla pipeline generation. The upper shows the specific details of constructing the novel OT reward. For illustration purposes, we add tag ($[e_i]$, $[s_i]$) for each event and sentence (i.e., not required in reality).

$$Lp = -\sum_{k=1}^{|\hat{S}_i|} (\gamma_1 logp(\hat{S}_{i,k}|S_i^e, \hat{S}_{i,<k}) + \quad (4)$$
$$\gamma_2 logp(\hat{S}_{i,k}|S_i^t, \hat{S}_{i,<k}))$$

where $\gamma_1$ and $\gamma_2$ are weight factors, varying with the pretraining process.

### 3.3 Optimal Transport Reward

An apparent problem with the planning-based method is sketch model can't adjust with story model as the sketch generation is non-differentiable. This leads to sketch model not knowing how the generated sketch affects the final story. To overcome this barrier, we design an RL algorithm with novel OT reward, further improving the generated event quality.

We briefly introduce how to utilize RL to optimize sketch model. Particularly, a reward $\hat{R}$ is constructed based on the feedback from story model. Later, the policy gradient (Sutton et al., 1999) is adopted to optimize sketch model through maximizing the expected reward $E_\alpha[\hat{R}_i]$ in (5). The gradient of sketch model can be represented as (6) according to the policy gradient theorem, which can be approximated with the sampling techniques.

$$E_\alpha[\hat{R}_i] = E[\hat{R}_i \cdot log(p(\hat{S}_i|x_i^t, \alpha)] \quad (5)$$
$$\nabla J(\alpha) = E[\hat{R}_i \cdot \nabla logp(\hat{S}_i|x_i^t, \alpha)] \quad (6)$$

Therefore, the core idea is to design an effective reward $\hat{R}_i$ that guides sketch model to generate better sketch. A naive approach is regarding negative sentence loss as the corresponding event reward. Because low-quality event in sketch will make it harder for story model to reconstruct the original sentence in story, leading to higher sentence loss and smaller event reward. However, the mappings between each event $e_i$ in sketch and the sentence $s_j$ in story modeled by the naive approach are one-to-one as the reward for $e_i$ is only determined by $s_i$. But they should be one-to-many as an event may contribute multiple sentences. To overcome this barrier, we design a novel OT reward.

As shown in the upper of Figure 3, we extract the event semantics $H_e = \{h_{ei}\}_{i=1}^{l_e}$ and sentence semantics $H_s = \{h_{sj}\}_{j=1}^{l_s}$ by averaging the representations of the included tokens from the last hidden state of sketch model and story model. Here, $l_e$ and $l_s$ represent the number of events in sketch and sentences in story respectively. The sentence loss $L_s = \{L_{sk}\}_{k=1}^{l_s}$ is also computed by adding up the loss of the included tokens. Later, we view the process of unfolding sketch into story as moving the event semantic distribution to sentence semantic distribution, and employ OT to capture the mappings between them. Concretely, the cost matrix $C \in R^{l_e \times l_s}$ is first calculated based on the simi-

**Algorithm 1:** Training And Optimization

**Input:** $D^{train} = \{x_i^t, \hat{Y}_i\}_{i=1}^N$, $\eta$ weight factor for OT loss, $\alpha$, $\theta$ parameters of sketch model and story model.

1   pretrain sketch model based on (4)
2   **for** *sample* $P = \{x_i^t, \hat{Y}_i\} \in D^{train}$ **do**
3       Compute loss $L_\alpha$, $L_\theta$ by (2), (3).
4       Extract $H_e$, $H_s$, $L_s$ from the output
5       Compute cost matrix $C$ by (7)
6       Compute transport matrix $T$ by IPOT
7       Compute compact reward $R_i$ and OT Loss $L_{ot}$ by (8), (9)
8       Expand $R_i$ to get final reward $\hat{R}_i$
9       $\nabla J(\alpha) = \hat{R}_i \cdot \nabla log P(\hat{S}_i | x_i^t, \alpha)$
10      $\nabla \alpha = \nabla J(\alpha) + \eta \nabla L_{ot}$
11      $\nabla \theta = \nabla L_\theta + \eta \nabla L_{ot}$
12      optimize the entire model based on $\nabla \alpha$, $\nabla \theta$
13   **end**

larity between events and sentences measured by RBF kernel as (7).

$$c_{ij} = 1 - exp(\frac{-||h_{ei} - h_{sj}||^2}{2\beta^2}) \qquad (7)$$

It means the closer the semantics of $h_{ei}$ and $h_{sj}$, the smaller the transportation cost between them. Then, IPOT is adopted to compute the transport matrix (maps) $T \in R^{l_e \times l_s}$ based on $C$, $H_e$ and $H_s$. $T_{ij}$ can be understood as the semantic transportation or contribution from $e_i$ to $s_j$. After obtaining the mappings $T$, we construct the compact reward $R = \{r_i\}_{i=1}^{l_e}$ and compute the OT loss $L_{ot}$ as below:

$$R_i = -TL_s \qquad (8)$$
$$L_{ot} = trace(T^T C) \qquad (9)$$

where $r_i$ indicates the reward for $e_i$. Last, we expand each $r_i$ to make its dimension consistent with the number of tokens contained in the corresponding event to get the final reward $\hat{R}$. In this way, $r_i$ is not only determined by the sentence loss $s_i$, but also by other sentence loss. The reward intensity of $s_j$ to $e_i$ depends on the semantic contribution ($T_{ij}$) from $e_i$ to $s_j$ captured by OT. Consequently, each event reward comprehensively considers the feedback of all sentences, thus effectively measuring the quality of events in sketch. Furthermore, sketch model can perceive the generated event quality and understand event correlations through comparing the reward of different events, thereby adjusting itself accordingly. It facilitates sketch model to improve the quality of generated events. The entire flow is shown in Algorithm 1.

## 4 Experiments

In this section, we first introduce the datasets, the compared models, and the evaluation measures in the experiments. Then, we show the experimental results and provide a detailed analysis.

### 4.1 Datasets

To verify the effectiveness of our framework in generating both short and long stories, we choose ROCStories (ROC) (Mostafazadeh et al., 2016) and WritingPrompts (WP) (Fan et al., 2018) as benchmark datasets. For pretraining data, we adopt the event sequences provided by Lin et al. (2020) and further process it as follows. For each pairwise events, the event triggers and contexts are fed into ECONET (Han et al., 2020) to obtain their pairwise event order. Last, we repeat the operations in section 3.1 and obtain a total of 100k event sequences for pretraining. More details about the datasets and pretraining data are shown in appendix A.1.

### 4.2 Compared Models

**Baselines. (1) Bart** (Lewis et al., 2020) is a typical seq2seq model for natural language generation. We directly fine-tuned Bart-base on both datasets. **(2) Bart (planning)** adopted the planning-based method based on (1). **(3) TemporalBart** (Lin et al., 2020) is pretrained with temporal event ordering and event infilling tasks. We utilize its pre-trained weights to initialize our sketch model. **(4) Megatron** (Xu et al., 2020) fine-tuned GPT2 (Radford et al., 2019) and leveraged external knowledge bases to enrich the generated stories on ROCStories. **(5) ContentPlanning** (Goldfarb-Tarrant et al., 2020) adopted the planning-based Bart-large model on WritingPrompts. **(6) Flashback** (Han et al., 2022) utilized temporal prompt to generate flashback in place, which is the strongest baseline for our framework. We also compare with its three variants for fair, which are only **RL**, only **Pretrained** and the integration of RL and Pretrained (**PR**), respectively. For **RL**, it regarded the negative story loss as the final reward $\hat{R}$. For **Pretrained**, it adopted an autoregressive manner at pretraining stage. Note that Flashback and its variants all served event arguments as mask unit. **(7) ChatGPT** is selected as its powerful ability in natural language generation. More details about the baselines are in Appendix A.3.

**Ablation Variants.** In addition to the baselines, we present the ablation variants of our framework. **(1) Vanilla** performs vanilla pipeline generation as shown in the bottom of Figure 3. (excluding bidirectional pretraining) **(2) BP** adopts bidirectional pretraining of event blank infilling and event temporal order based on (1). **(3) OTRL** attaches the RL algorithm with OT reward based on (1).

| Models | ROCStories | | | | | WritingPrompts | | | | | |
|---|---|---|---|---|---|---|---|---|---|---|---|
| | PPL↓ | B-3↑ | $R_L$↑ | R-2↓ | D-4↑ | PPL↓ | B-3↑ | $R_L$↑ | R-3↓ | D-4↑ | Tks↑ |
| Bart | 20.24 | 4.98 | 19.11 | 47.78 | 62.44 | 31.15 | 0.57 | 9.28 | **20.92** | 59.37 | 148.6 |
| Bart (planning) | 27.30 | 5.13 | 19.29 | 49.77 | 61.80 | 31.04 | 0.67 | 9.43 | 23.40 | 60.27 | 160.2 |
| TemporalBart* | 24.65 | 5.01 | 19.12 | 50.62 | 62.13 | - | - | - | - | - | - |
| Flashback | 22.85 | 5.07 | 19.39 | 46.11 | 63.42 | 30.77 | 1.44 | 10.95 | 23.70 | 59.83 | 208.6 |
| + Pretrained* | 21.16 | 5.06 | 19.43 | 49.16 | 62.79 | 30.70 | 1.68 | 11.13 | 25.90 | 58.90 | 226.7 |
| + RL | 15.67 | 5.12 | 19.41 | 49.43 | 64.10 | 30.98 | 1.39 | 10.78 | 23.60 | 59.12 | 203.8 |
| PR* | 15.45 | 5.20 | 19.49 | 50.05 | 64.76 | 30.73 | 1.64 | 11.03 | 24.20 | 58.86 | 222.4 |
| Vanilla | 25.51 | 5.08 | 19.40 | 47.68 | 62.55 | 30.71 | 1.97 | 11.34 | 25.20 | 57.89 | 248.9 |
| + BP* | 25.35 | 5.06 | 19.43 | **46.02** | 64.21 | 30.62 | **2.10** | **11.51** | 24.80 | 59.99 | 255.7 |
| + OTRL | **14.73** | 5.27 | 19.61 | 49.01 | 65.09 | **30.41** | 2.09 | **11.51** | 26.10 | 60.15 | **256.9** |
| BPOT* | 14.85 | **5.31** | **19.64** | 48.17 | **65.51** | 30.54 | 2.04 | 11.43 | 24.60 | **60.39** | 253.9 |

Table 2: Automatic evaluation results. * represents that it includes pretraining stage. The bottom shows the results of our framework and its ablation variants. Best in bold, the runner-up with an underline.

## 4.3 Evaluation Measures

**Automatic Evaluation** We use the following automatic metrics to evaluate stories. **PPL** represents the model's perplexity of the stories. **Repeat-n (R-n)** (Shao et al., 2019) reflects the redundancy of the stories by computing the ratio of the stories that repeat at least one n-gram. **Distinct-n (D-n)** measures the diversity of the stories by computing the ratio of n-gram types to all generated n-grams. **Tks** is the average length of the stories. We also report standard **BLEU-3 (B-3)** (Papineni et al., 2002) and $\mathbf{ROUGE}_L(\mathbf{R}_L)$ (Lin, 2004).

**Manual Evaluation** We conduct a manual evaluation on ROCStories to verify whether the storytelling system can **generate the high-quality events that balance the logic and narrative order in generating stories with flashbacks.** Specifically, we randomly choose 100 stories that have *<after>* in test sets. For each story, we obtain six versions which are generated by the ablation variants of our framework and three strong baselines. Then, the annotators are required to evaluate stories on three aspects: **Narrative order**, **Coherence** and **Overall**. For Narrative order, we ask them to label all pairwise event orders in each story. Then the ratio of each narrative order is calculated, and we further compute the entropy to represent **Narrative Order Diversity (NOD)**. Meanwhile, **Narrative Order Accuracy (NOA)** is measured by calculating the ratio of the annotated results that are consistent with the given temporal prompts. For **Coherence**, we have the annotators rate each story (1-5) according to the inter-sentence logic and whether the generated story deviates from the given input. For **Overall**, the annotators are required to rank

stories based on the overall quality of the story and their preferences. Also, they can line up two stories together if they are fairly similar in quality. For each set of stories, we have 5 workers to annotate it. Appendix A.4 shows more details on automatic and manual evaluation.

## 4.4 Experimental Results

**Automatic Evaluation Results** are reported in Table 2. BPOT surpasses other baselines in all metrics except for repetition, which indicates the generated stories are more fluent and diverse, overlapping more with the reference stories. For **repetition**, our model outperforms other baselines under similar configurations (e.g., RL vs OTRL) on ROCStories, while showing slightly worse results on WritingPrompts, probably because of the much longer generated stories. Concretely, when compared to the strongest baselines (Flashback and its variants), Vanilla performs worse than Flashback on ROCStroies. It could be that Flashback serves event arguments as mask unit, which helps learn the dependencies between event arguments. Instead, Vanilla is better than Flashback on WritingPompts, especially in **Tks**. It may be because the stories in WritingPrompts are much longer. Thus, the dependencies between event arguments are complex

| Models | B-3↑ | $R_L$↑ | R-2↓ | R-3↓ | D-4↑ | Tks↑ |
|---|---|---|---|---|---|---|
| Megatron (ROC) | 2.57 | 19.29 | 60.75 | - | 85.42 | - |
| ContentPlanning (WP) | 3.46 | 14.40 | - | 95.60 | 78.16 | 252.3 |
| OTRL (ROC) | **5.27** | **19.61** | 49.01 | - | 65.09 | - |
| OTRL (Bart-large) (WP) | 3.26 | 13.21 | - | **30.80** | 61.29 | **323.4** |

Table 3: Comparison results on two datasets. PPL is missing as it is not reported in original paper.

| Models | NOD ↑ | NOA ↑ | Coherence ↑ | Overall↓ |
|--------|-------|-------|-------------|----------|
| TemporalBart* | 0.913 | 0.903 | 2.543 | 3.398 |
| PR* | 1.037 | 0.913 | 2.923 | 2.975 |
| Vanilla | 0.954 | 0.900 | 2.400 | 3.555 |
| OTRL | 0.985 | 0.915 | 3.425 | 2.590 |
| ChatGPT | 0.474 | 0.745 | **4.403** | **1.530** |
| BPOT* | **1.115** | **0.958** | 4.113 | 1.820 |

Table 4: Manual evaluation results on ROCStories. Best in bold, the runner-up with an underline.

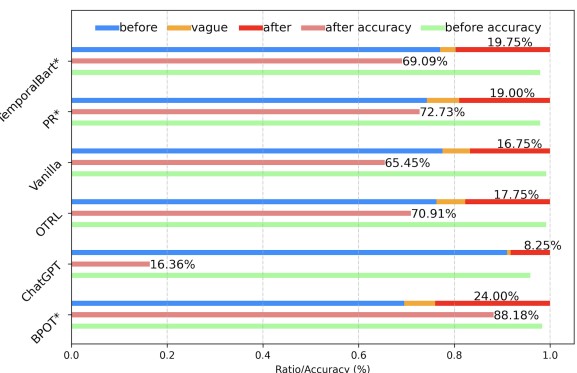

Figure 4: Detailed statistics on the annotated results of narrative order. For each set, the first line shows the ratio of different narrative orders, the second and third line shows the **NOA** on *<after>* and *<before>* respectively.

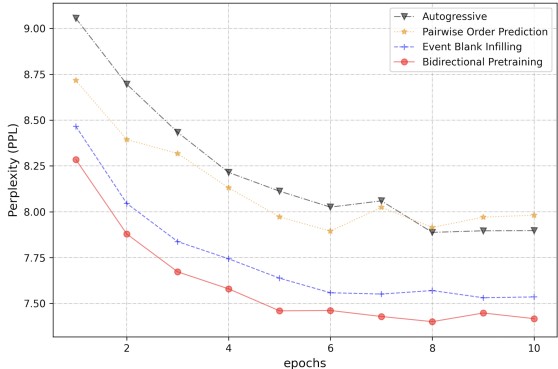

Figure 5: The Perplexity (PPL) of different strategies in pretraining stage.

| Models | PPL↓ | B-3↑ | $R_L$↑ | R-2↓ | D-4↑ |
|--------|------|------|--------|------|------|
| Autoregressive | 15.35 | 5.22 | 19.53 | 49.06 | 64.97 |
| Event Blank Infilling | 15.34 | 5.23 | 19.55 | 49.94 | 65.11 |
| Pairwise Order Prediction | 15.04 | 5.26 | 19.48 | 48.73 | 64.46 |
| BPOT* | **14.85** | **5.31** | **19.64** | 48.17 | **65.51** |
| RL | 15.87 | 5.16 | 19.40 | 50.31 | 62.76 |
| Naive-RL | 15.37 | 5.12 | 19.48 | **46.74** | 63.57 |
| OTRL | **14.73** | 5.27 | 19.61 | 49.01 | **65.09** |

Table 5: The results of comparative experiments. The upper represents different pretraining strategies combined with OTRL. The bottom represents different rewards, the specific difference is shown in Appendix 8.

and hard to learn, while serving event as mask unit makes it easier to learn event correlations, thereby generating longer stories. When attaching OT reward, OTRL outperforms all variants of Flashback. Besides, although our pretraining data is about 0.1 of Pretrained (1 million), BP performs better, especially in diversity and repetition. BP achieves a growth of 1.66 and 2.1 in **Distinction** and a decline of 1.66 and 0.4 in **Repetition** on two datasets, while Pretrained are $-0.63$, $-0.93$ and $-3.05$, $-2.2$ respectively. The superior performance on two datasets demonstrates the effectiveness of bidirectional pretraining and OT reward.

Moreover, since the experimental settings of Megatron (Xu et al., 2020) and ContentPlanning (Goldfarb-Tarrant et al., 2020) are different from other baselines, we conducted separate experiments to compare with them. The results are shown in Table 3. Megatron achieves better results in diversity as it served GPT2 as the backbone and leveraged external knowledge bases to insert novel words into stories. Content Planning outperforms ours in some metrics because it used more training data and designed a series of classifiers to refine sketch. **Manual Evaluation Results** are shown in Table 4. More detailed statistics on the annotated results of

narrative order are shown in Figure 4. We can find that the dominant narrative order of the generated story is straightforward (*<before>*), which is consistent with human writing habits and the findings in Ning et al. (2018). BPOT performs best in **NOD** (1.115) and **NOA** (0.958). It proves that inversely inferring pairwise order through bidirectional pretraining facilitates model to be more better aware of *<after>*, thus reversing narrative order to generate flashback (highest after ratio 24% and highest after accuracy 88.18%). Moreover, OTRL greatly boosts the coherence and overall quality. We believe that the reason is that OT reward effectively measures the quality of the generated events. Therefore, sketch model can adjust with reward to generate higher-quality events. Besides, the bidirectional pretraining also improves the coherence and overall quality. We believe that it is because event correlations encoded by bidirectional pretraining facilitate sketch model to plan reasonable event logic, thus alleviating the logical incoherence. Particularly, although ChatGPT performs better in coherence and overall due to its huge number of parameters and rich training corpus, it is difficult for ChatGPT to break the dominant narrative order *<before>* and

| | |
|---|---|
| **Input prompt** | [s1] terry was in bed sleeping. 
 *<before><before><after><before>* |
| PR* | [s2] suddenly, he heard a loud crash. 
 [s3] he jumped out of bed to see what it was. 
 [s4] *it was a car accident.* 
 [s5] terry rushed to the hospital. |
| ChatGPT | [s2] suddenly, the alarm clock blared. 
 [s3] he turned it off and sat up. 
 [s4] he realized he wad late for an important meeting. 
 [s5] frantically, he rushed to get dressed and left the house. |
| BPOT* | [s2] suddenly, he heard a loud bang. 
 [s3] he looked out the window and saw a bear. 
 [s4] *the bear had attacked terry's house.* 
 [s5] terry ran outside and called the police. |
| **Input prompt** | [s1] a friend of mine just broke up. 
 *<after><before><after><before>* |
| PR* | [s2] *she had a boyfriend who was cheating on her.* 
 [s3] i tried to talk to him, but he wouldn't talk to me. 
 [s4] i asked him why he cheated on me. 
 [s5] he told me that he didn't want to be friends with me. |
| ChatGPT | [s2] i consoled him with ice cream and movies. 
 [s3] later, he admitted it was for the best. 
 [s4] after a few months, he found someone new. 
 [s5] now, he's happier than ever before. |
| BPOT* | [s2] *she had a boyfriend who was cheating on her.* 
 [s3] i asked her why. 
 [s3] *she told me that he cheated on her with another woman.* 
 [s5] i felt bad for her. |

Table 6: The generated stories from different models. The brown and the yellow represent the correct and failed generated flashbacks respectively. The red represents the logical incoherence.

follow *<after>* to generate flashback. It is reflected in the poor performance of **NOD** (0.474), lowest after ratio 8.25% and **NOA** (0.745) , lowest after accuracy 16.36%.

**Analysis of BP and OTRL** To further verify the effectiveness of BP and OTRL, we conduct comparative experiments on ROCStories. As shown in Figure 5 and Table 5, we compare various pretraining strategies and rewards under the same experimental settings (serving event as mask unit). We can find that BP achieves the lowest **PPL** during pretraining stage. After combing with OTRL, BPOT outperforms other control groups in all metrics. It proves that the bidirectional reasoning is superior to unidirectional reasoning and the autoregressive manner. Besides, when compared to RL and Naive-RL, OTRL performs best. It demonstrates the effectiveness of measuring the event quality in fine-grained way and further considering the one-to-many mappings between the generated events and the sentences in the story. Both comparative experiments verify the superiority of bidirectional pretraining and OT reward.

### 4.5 Case Study

Two cases are shown in Figure 7. We find that ChatGPT is almost able to generate reasonable event logic. However, it fails to follow the *<after>* and reverse narrative order to generate flashback in place. Moreover, for simple situations with one *<after>*,

PR sometimes generates correct flashback. But it is faced with logical incoherence. Seeing the upper example, "terry rushed to the hospital" follows "it was a car accident". But the accident is not related to terry so that there is no need for him to rush to the hospital. Instead, BPOT generates the correct flashback ("had attacked terry's house." occurs in the past and explains why "hearing a loud bang") and maintains the reasonable event logic. Besides, for complex situations with multiple *<after>*, PR faces the problems of inefficient prompt and deteriorating logic. As shown in the bottom example, it fails to generate flashback with second *<after>* and the conflicting relationship with that man has shifted from my friend to me. In contrast, BPOT generates two correct flashbacks ("had a boyfriend who was cheating on her" occurs in the past and explains "broke up", "cheated on her" occurs in the past and explains "asked her why"). Both cases demonstrate the effectiveness of our framework in generating logically coherent stories with flashbacks. More cases are shown in Appendix.

## 5 Related Work

**Story Generation** was first approached by symbolic and logical planning (PÉrez and Sharples, 2001; Martens et al., 2014). Recently, significant progress has been made in applying deep neural networks in story generation. Fan et al. (2018); Mao et al. (2019) reused a seq2seq model to translate the prompt into a story. Xu et al. (2018); Yao et al. (2019); Martin et al. (2018) introduced the planning-based method, namely planning sketch prior to writing story. Then, numerous works were devoted to designing the format of sketch (Fan et al., 2019; Chen et al., 2020) and enriching sketch with external knowledge bases (Tan et al., 2021; Xu et al., 2020). Moreover, researchers explored controllable story generation through fine-grained control of sketch, such as writing style (Kong et al., 2021), protagonist's personality (Zhang et al., 2022). However, they faced the problem of logical incoherence and rarely considered the crucial role of narrative order in story generation. Although Han et al. (2022) proposed the temporal prompt to generate flashback, it faced the problems of inefficient prompt and deteriorating logic in some cases. In this work, we focus on generating logically coherent stories with flashbacks.

**Event Correlations and Temporal Order** have been proven useful in many event-related tasks

([Chen et al., 2023](); [Liu et al., 2022]()). Recent works paid attention to utilizing pretraining to encode event correlations and event temporal order. [Han et al. (2020)]() learned to identify the temporal relationship between events through masking event triggers and temporal indicators. [Lin et al. (2020)](); [Zhou et al. (2022a)]() explored temporal event ordering and event infilling tasks for mining temporal knowledge. [Zhou et al. (2022b,c)]() adopted event-level pretraining with contrastive learning to capture event correlations. However, they rarely involve mutual reasoning between pairwise event and event order. In this work, we jointly encode event correlations and pairwise event order through the bidirectional reasoning between event and order.

## 6 Conclusions

In this paper, we propose a narrative order aware framework BPOT for story generation, which presents a bidirectional pretrained model to encode event correlations and pairwise event order. We also design an RL algorithm with novel OT reward to further improve the generated event quality in the fine-tuning stage. Both automatic and manual evaluation results demonstrate the superiority of our framework in generating logically coherent stories with flashbacks. In the future, we will explore how to control the narrative order of long texts (paragraphs) or other narrative modalities (video).

## Limitations

The performance of our proposed framework is related to the used pretrained language model (PLM). Applying our proposed framework to stronger PLM may lead to further improvements. Besides, the temporal prompts are obtained by ECONET through majority voting with different random seeds. This inevitably introduces some noise in the data, possibly affecting the final performance.

## Acknowledgements

This research was funded by the National Natural Science Foundation of China (62206267). We sincerely thank Shiyao Yan, Changyuan Tian, and Wei Jia for their constructive collaboration.

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

# A Appendix

## A.1 Details on Datasets and Pretraining data

**ROCStories** (Mostafazadeh et al., 2016) It contains 98,162 five-sentence short stories. The average length of the story is about 42 words. Following (Xu et al., 2020; Han et al., 2022) we split the data into 88,344/4.908/4,909 for train/validation/test sets.

**WritingPrompts** (Fan et al., 2018) It contains 30,335 pairs of prompts and stories. The average length of the story is over 700 words. Thus, the stories in WritingPrompts are much longer than those in ROCStories. Following (Han et al., 2022), we select stories with a maximum of 500 words, leading to a total number of 96,488 training and 5,784 validation prompt-story pairs, respectively. For the test set, we utilize the 1000 prompt-story pairs provided by the compared baseline (Goldfarb-Tarrant et al., 2020). The temporal prompts in both datasets are provided by Han et al. (2022).

**Pretraining data** For pretraining data, we utilize the event sequences provided by Lin et al. (2020). It is originally used to pretrain with the tasks of temporal event ordering and event infilling. Thus, the event sequences are full of the knowledge of event correlations and event temporal order. We further process it as follows. For each event sequence, we segment all its pairwise adjacent events. For each pairwise event, we separately restore their original sentences and detect their event triggers by Han et al. (2019). Then we feed their event triggers and contexts into three ECONET (Han et al., 2020) with different random seeds to get pairwise event order. We only adopt the result when the event orders judged by the three models are the same. Otherwise, we label its pairwise event order as *<vague>*, which means arbitrary event order. Note that there may be multiple results of semantic role labeling for a sentence in datasets, we choose the results whose event trigger is the ROOT node in the syntactic analysis as our event. By doing so,

we finally obtain a total of 100k event sequences as our golden sketches for pretraining.

## A.2 Implementation Details

Unless specifically mentioned, we use the pre-trained Bart-base [*] as our sketch model and story model. The initial vocabulary of BART contains 50265 tokens. If the compared model which could be baselines or ablation variants of our framework adopts the planning-based method, we add an event ending (*<eoe>*) token. If it inserts temporal prompts into sketch, we add three additional narrative order tokens (*<before>*, *<after>*, *<vague>*). On ROCStories, we serve the event in the first sentence of each story as the input event. On Writing-Prompts, we serve the prompt as the input event. For ROCStories, the hyper-parameters are learning rate: $5e^{-5}$; batch size: 10; $\beta$: 0.7; gradient accumulation: 1. To fairly compare with (Han et al., 2022), we use three random seeds (5, 9998, 20016) and report the average performance of all models evaluated on ROCStories. For WritingPrompts, the hyper-parameters are learning rate $2e^{-5}$; batch size: 64; $\beta$:0.7, gradient accumulation: 8. For both datasets, we fine-tune on a single Tesla V100 GPU with 32G memory. When compared with the baseline ContentPlanning (Goldfarb-Tarrant et al., 2020), we fine-tune Bart-large on a single A100 GPU with 80G memory. The training time for per epoch on ROCStories and WritingPrompts is 2.5-3.5 hours and 24 hours respectively. We train our framework and its ablation variants for 10 epochs and save the model with the best evaluation perplexity. For pretraining, hyper-parameters are learning rate $1e^{-5}$; batch size: 10; gradient accumulation: 10. We also pretrain 10 epochs and save the model with the best evaluation perplexity. The weight factor $\eta_1$, $\eta_2$ during pretraining are computed as follows:

$$\eta_1 = 3 - 2 * (global\ step/total\ step) \quad (10)$$
$$\eta_2 = 1 + 2 * (global\ step/total\ step) \quad (11)$$

where the global step represents the current number of iterations that have been updated and the total step represents the total number of iterations to be updated. It means that the influence of event blank infilling will gradually increase during pretraining, while the influence of pairwise order prediction will gradually decrease. It is because we hope

_______________

[*]https://huggingface.co/facebook/bart-base/tree/main

model can focus more on pairwise order prediction in the early stage of pretraining, then utilizing the awareness of event temporal order to generate better events in event blank infilling.

## A.3 Details on Baselines

**Bart:** (Lewis et al., 2020) It is a typical seq2seq model for natural language generation. We fine-tune it on both datasets. We directly serve the first sentence of story on ROCStories and prompt of story on WritingPrompts as the input, and the entire story as labels to fine-tune the model.

**Bart (planning):** It adopts the planning-based method based on Bart. We first use sketch model to generate sketch and then use story model to generate story given on the generated sketch. The detailed process is consistent with the vanilla pipeline generation at the bottom of Figure 3 except that we replace all temporal prompts with *<eoe>*.

**TemporalBart:** (Lin et al., 2020) It is designed with the tasks of temporal event ordering and event infilling. We pick it as it implicitly encodes event correlations and event temporal order. Note that the temporal prompt is not used during pretraining. Concretely, we use its pre-trained weights to initialize the sketch model and then adopt the workflow of Flashback (Han et al., 2022) to complete the subsequent fine-tuning.

**Megatron:** (Xu et al., 2020) It used GPT2(Radford et al., 2019) and leveraged external knowledge bases to generate the commonsense stories. We picked it as it outperforms previous systems (Guan et al., 2020) on ROCStories. Due to its operation of delexicalization that replaces names and entities with [MALE], [FEMALE], and [NEUTRAL], when compared with it, we strive to recover its original entities and name.

**ContentPlanning:** (Goldfarb-Tarrant et al., 2020). It adopted the planning-based method on Writing-Prompts. As reported in Han et al. (2022), our final training data is about two-thirds of theirs. Moreover, they do not adopt end-to-end training as they additionally design a series of classifiers to refine the sketch.

**Flashback:** (Han et al., 2022). It first considered the crucial role of narrative order in story generation and proposed temporal prompt to generate flashback. It is the strongest baseline for our work. The detailed process is consistent with the bottom of Figure 3 except it served event arguments as mask unit. Concretely, each masked

event in input is represented as "<mask> ; <mask> ; <mask> ; <Temporal Prompt>" rather than "<mask><Temporal Prompt>" in our framework. We serve event as mask unit as we deem that it facilitates model to directly learn event correlations, especially in long stories. The results of automatic evaluation on WritingPrompts also demonstrate it. To further verify the effectiveness of our methods, we compare with its three variants. The only **RL** simply regards the negative story loss as reward $\hat{R}$, which means that it returns an identical reward for all events in the generated sketch. The **Pretrained** used 1 million event sequences which are ten times for our pretraining data. Moreover, it adopts autoregerssive manner during pretrianing stage. The autoregressive manner represents that it unidirectionally reasons all event arguments from left to right. The **PR** is the integration of **RL** and **Pretrained**, which means that it uses the pre-trained weights to initialize sketch model and combines with RL in the fine-tuning stage.

**ChatGPT:** we pick it because of its powerful ability in NLP. We only compare with it on ROCStories. Because the stories in WritingPrompts are much longer and contain dialogue or short phrases without events, which makes it hard for annotators to judge the order of pairwise event. Referring to the tutorial [†], we design the instruction for ChatGPT as shown in Figure 6. Note that we prompt ChatGPT to generate stories no more than 48 words as the average length of the story on ROCStories is about 42. If the length is not constrained, the stories generated by ChatGPT are much longer than 42 words, which is adverse to fair comparison. Moreover, although we hint the story should be five-sentence, ChatGPT sometimes generates more than five sentences. In this case, we only choose the first five sentences as the final story.

## A.4 Details on Evaluation Measures

**Automatic Evaluation Measures:** We find that our models can achieve nearly 0 in **Repeat-3** on ROCStories and in **Repeat-4** on WritingPrompts, which is in line with the findings in Han et al. (2022). Therefore, we report **Repeat-2** on ROCStories and **Repeat-3** on WritingPrompts. For **Tks**, we only report on WritingPrompts. Because the stories on ROCStories are relatively short so that the generated stories are almost the same length through full

Figure 6: The instructions for ChatGPT on ROCStories.

training. For WritingPrompts, it is hard for models to generate such long stories in training sets due to the limitation of the model's capacity. Therefore, the **Tks** can reflect the performance of the model to some degree.

**Manual Evaluation Measures:** We randomly sample 100 stories from the test sets of ROCStories. For each story, we get 6 versions which are generated by different storytelling systems. Each set of stories is evaluated by 5 annotators on three aspects: **Narrative Order**, **Coherence**, **Overall**. It should be noted that all annotators have linguistic backgrounds and the order of storytelling systems in each set is shuffled. The specific instructions are shown in Figure 7 and Figure 9. Specifically, we clearly introduce the definition of the task and the evaluation method for each metric. Moreover, we provide an example with the detailed explanation for the corresponding task. For each set of stories, we require four annotators to rate coherence and rank overall. Then, we average their values and obtain the final results reported in the manual evaluations. We also ask one annotator to label the pairwise event order to get the narrative order.

## A.5 Comparison of Reward

The comparison of various rewards is shown in Figure 8. The **RL** in (Han et al., 2022) is constructed at the story-level, which means that all

**Instruction for [task1: judge narrative order]**

In this task, you will be given a set of stories. It contains six five-sentence stories generated by different storytelling systems along with the same input, denoted as A, B, C, D, E, F. You are required to judge the narrative order of each story in set. The narrative order of the story is reflected on pairwise event order in story. Therefore, you are required to label all pairwise event orders for each story. The types of event order include "before", "after", "vague". Specifically, if event1 occurs earlier than event2, then the event order between them is "before". If event2 occurs later than event2, then the event order between them is "after". If you are not sure about the order between pairwise event, then you label "vague". Here is a detailed reference example with explanations. Note that the sentence is separated by ".", "?" or "!".

Referenced Story:
dominic was known to be a trouble maker. one day, he decided to break into a neighbor's house. he broke into the house and stole all of his things. dominic's parents were very upset. They grounded him for a month.

[sentence 1&2] Explanation: The event in sentence1 is "was known to be a trouble maker.", the event in sentence2 is "decided to break into a neighbor's house". We can obtain from the time indicator "one day" that event1 started earlier than event2.

[sentence 2&3] Explanation: The event in sentence2 is "decided to break into a neighbor's house", the events in sentence3 is "broke into the house" and "stole all of his things". It is clear that "decider to break into" started earlier than "broke into" and "stole all of his things".

[sentence 3&4] Explanation: The evens in sentence3 is "broke into the house" and "stole all of his things". The event in sentence4 is "were very upset.". It is obvious that "were very upset" is caused by "broke into the house" and "stole all of his things". Thus "broke into the house" started earlier.

[sentence 4&5] Explanation: The event in sentence4 is "were very upset.". The event in sentence5 is "grounded him". It is clear that "were very upset" started earlier than "grounded him". Thus "were very upset" started earlier.

Figure 7: Instruction for narrative order

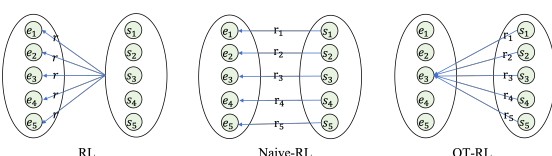

Figure 8: Comparison of different rewards in reinforcement learning

events in the generated sketch receive the identical reward. The reward is regarded as the negative story loss. Therefore, it cannot measure the quality of the generated events because there is no difference between their rewards. The **Naive-RL** is constructed at the event-level, which means that different events in the generated sketch receive the different rewards. The reward is regarded as the negative sentence loss. Therefore, the sketch model can perceive the quality of generated event through comparing the reward of different events. However, the mappings between the generated events and the sentences in the story are modeled by one-to-one. But they actually should be one-to-many as an event may influence multiple sentences in story. Therefore, it can't effectively measure the event quality as the mappings are inaccurate. The **OT-RL** is constructed at the event-level. It also means that different events in the generated sketch receive the different rewards. The reward is regarded as the weighted sum of sentence loss. The weights are the mappings between the generated events and the sentences in the story, which are captured by OT.

In this task, you will be given a set of stories. It contains six five-sentence stories generated by different storytelling systems along with the same input, denoted as A, B, C, D, E, F. You are required to rate (1-5) the coherence of each story in set. The coherence of the story is reflected on the logical correlation of inter-sentence and whether the generated story deviates from the given input. You are also required to rank (1-6) the stories in set according their overall quality. Actually, it is difficult for you to make a quantitative judgment on the overall quality of the story. We hope you can rank the quality of the story based on your own preference after considering the fluency, interest of the story. If you deem that the quality of two stories is fairly similar, you can assign them the same ranking. Here is a detailed reference example with explanations.

System A: dominic was known to be a trouble maker. one day, he decided to go to the store to buy a new bike. when he got there, he realized he didn't have his bike. he went back home and changed his tire. now, dominic doesn't have a bike.
System B: dominic was known to be a trouble maker. one day, he decided to get into a fight with a girl. the girl was in trouble with the police. the police arrested dominic. now, dominic doesn't get in trouble anymore.
System C: dominic was known to be a trouble maker. one day, he decided to make a video. he made a video of himself throwing a rock at a neighbor's window. the neighbor saw the video and called the police. the police arrested dominic for making the video.
System D: dominic was known to be a trouble maker. one day, he decided to break into a neighbor's house. he broke into the house and stole all of his things. dominic's parents were very upset. they grounded him for a month.
System E: dominic was known to be a trouble maker. one day, he decided to go to the police station. the police officer told him that he had a warrant for his arrest. dominic went to jail. he was sentenced to community service.
System F: dominic was known to be a troublemaker. one day, he found an old map hidden in his attic. intrigued, he embarked on a treasure hunt. he followed the clues, digging up the backyard. finally, he discovered a chest filled with ancient artifacts.

Figure 9: Instruction for coherence and overall

| Input prompt | [s1] my friend joe has kidney problems. *<before><after><vague><vague>* |
|---|---|
| PR* | [s2] he is trying to get better. |
| | [s3] he went to the doctor for a checkup. |
| | [s4] *the doctor told joe he had kidney stones.* |
| | [s5] joe said it was not a problem. |
| ChatGPT | [s2] he undergoes a successful transplant. |
| | [s3] he feels like a new person. |
| | [s4] he starts volunteering at the hospital. |
| | [s5] he inspires others to give thegift of life. |
| BPOT* | [s2] he went to the hospital to get his kidney checked out. |
| | [s3] *the doctor said joe had kidney stones.* |
| | [s4] *joe was in a lot of pain.* |
| | [s5] he had surgery to remove the stones. |
| **Input prompt** | [s1] kim had a burning sensation in her bladder. *<before><after><before><before>* |
| PR* | [s2] she went to the bathroom to find out what it was. |
| | [s3] she realized she had flushed the toilet. |
| | [s4] she called the plumber. |
| | [s5] he came and fixed the toilet for her. |
| ChatGPT | [s2] she rushed to the restroom, but it was occupied. |
| | [s3] the discomfort grew unbearable. |
| | [s4] finally, she found relief and sighed in relief. |
| | [s5] kim made a mental note to take care of her health. |
| BPOT* | [s2] she went to the bathroom to see what was wrong. |
| | [s3] *she realized that she had a clogged bladder.* |
| | [s4] kim had to go to the emergency room. |
| | [s5] the doctor gave her some medication to help her. |
| **Input prompt** | [s1] tom was depressed. *<after><before><vague><before>* |
| PR* | [s2] *he didn't have much money.* |
| | [s3] he decided to buy a lottery ticket. |
| | [s4] he spent a lot of money. |
| | [s5] tom won the jackpot. |
| ChatGPT | [s2] *he didn't feel like doing anything.* |
| | [s3] he lay in bed all day, scrolling through social media. |
| | [s4] as the day went on, he felt worse. |
| | [s5] he decided to call a friend for help. |
| BPOT* | [s2] *he didn't have anything to cheer him up.* |
| | [s3] he decided to go to the doctor. |
| | [s4] the doctor gave him a prescription for antidepressants. |
| | [s5] tom felt much better. |

Table 7: More case studies on ROCStories. The brown and the yellow represent correct and failed flashbacks. The red represents the logical incoherence.