# OpenReview forum: "Narrative Order Aware Story Generation via Bidirectional Pretraining Model with Optimal Transport Reward"
_EMNLP/2023/Conference — EMNLP 2023 Findings_

### Official Review · Reviewer_tdcS · 2023-07-22

**Typos Grammar Style And Presentation Improvements:** Line 142, zh
**Soundness:** 4

**Excitement:**

3: Ambivalent: It has merits (e.g., it reports state-of-the-art results, the idea is nice), but there are key weaknesses (e.g., it describes incremental work), and it can significantly benefit from another round of revision. However, I won't object to accepting it if my co-reviewers champion it.

**Paper Topic And Main Contributions:**

This paper proposes a narrative order aware framework BPOT for story generation, which presents a bidirectional pretrained model to encode event correlations and pairwise event order. They also designed a RL algorithm with novel OT reward to further improve the generated event quality in fine-tuning stage. Both automatic and manual evaluation results demonstrate the superiority of BPOT in generating logically coherent story with flashback.

**Questions For The Authors:**

1. How to compute PPL for planning-based models? And how to compare PPL of different models with different formats (e.g., with or without temporal prompts)
2. Can you clarify the intuition and motivation of the OT reward? I am not familiar with IPOT. It is better to let readers know how to compute T.
3. When the sketch is model-generated, how do you compute the story loss?
4. What is the relation between Eq.8 and Eq.9? How are they used, and what are they used for?

**Reasons To Accept:**

1. The reward design is novel
2. The experiment is sufficient and convincing.

**Reasons To Reject:**

1. The method part is difficult to follow, including the wording, structure, and graphs.
2. Zero-shot ChatGPT is still very strong. So I am suspectible of the research significance of story generation.

**Reproducibility:**

3: Could reproduce the results with some difficulty. The settings of parameters are underspecified or subjectively determined; the training/evaluation data are not widely available.

**Reviewer Confidence:**

4: Quite sure. I tried to check the important points carefully. It's unlikely, though conceivable, that I missed something that should affect my ratings.

---

> ### Author Rebuttal · Authors · 2023-08-28
>
> Thank you very much for your positive feedback and constructive comments.
>
> (1) Regarding the difficulty of following the method and reproducibility.
> ​	​	Res: Thank you for your detailed comments. Actually, the obscurity of the concept of "narrative order" itself makes it more difficult to understand the method. With that in mind, we have made great efforts to clearly express our approach. For example, the section "Preliminary" is set up to help understand the basis of our method. Each core design of our method is equipped with a figure to explain (Figure 2 for bidirectional pretraining, Figure 3 and Figure 9 for optimal transport reward). We also provide pseudo code to illustrate the entire algorithm. In the camera-ready version, we will further polish our paper to make it easier to understand. We will also make the code publicly available for comprehension and reproduction.
>
> (2) Regarding the research significance of story generation.
> ​	​	Res: Thank you for your detailed comments. It is well known that ChatGPT is a very powerful natural language generation tool. However, recent studies have shown that current ChatGPT performed poorly in event temporal reasoning [1] [2]. And its poor performance in narrative order story generation reported in our paper (0.474 in Narrative Order Diversity, 0.745 in Narrative Order Accuracy) also supports this view. This is because narrative order story generation requires the model to have temporal reasoning capabilities to understand the pairwise event order. Therefore, current ChatGPT might perform well in general story generation. However, it still has a gap in the performance of fine-grained controllable story generation that combines narrative order, causality, sentiment, etc. Thus, we believe that story generation is still worth researching. To a certain extent, it can also reveal the shortcomings of LLM and make improvements.
> [1] Chatgpt evaluation on sentence level relations: A focus on temporal, causal, and discourse relations.  2023 arXiv
> [2] Towards Benchmarking and Improving the Temporal Reasoning Capability of Large Language Models.  2023 ACL
>
> (3) Regarding the computation and comparison of PPL.
> ​	​	Res: Thank you for your detailed comments. PPL in this paper represents the model's perplexity of the golden stories. Formally, let $S_i$ , $Y_i$ denote the $i$-th generated sketch and $i$-th golden story in planning-based model, respectively. Then, the perplexity of $i$-th golden story $PPL_i$ can be computed as below:  $$PPL_i = exp(-\sum_{k=1}^{|\hat{Y_i}|}logp(\hat{Y_{i,k}}|S_i,\hat{Y_{i,<k}}))$$
> $p(\hat{Y_{i,k}}|S_i,\hat{Y_{i,<k}})$ is the probability of the golden token $\hat{Y_{i,k}}$, which is computed by the last hidden state of  bart decoder. The overall PPL of the golden stories in testset is the average of all $PPL_i$, which can be represented as: $PPL = \frac{1}{N}\sum_{i=1}^{N}PPL_i$, where N is the number of the golden stories in testset. Note that for different models, their formats of $S_i$ may be different (with or without temporal prompts), but their target output remains consistent (pure golden story $Y_i$ without temporal prompts). As a result, we can use the same method to calculate PPL for different models and directly compare them.
>
> (4) Regarding the intuition and motivation of OT reward.
> ​	​	Res: Thank you for your detailed comments. The intuition and motivation of OT reward is to model the one-to-many mappings between the generated events and the sentences in story so as to effectively measure the quality of the generated events. Specifically, the model is optimized with the reward of RL in fine-tuning stage. And we hope that lower event quality leads to lower reward. Then the reward can guide model to generate higher-quality event by punishing low-quality event that represents low reward.  Thus, effectively measuring the event quality is critical as the optimization can be misleading if the reward is not accurate. However, existing rewards can't effectively measure the generated event quality. Briefly, they either fail to differentiate the quality of generated events (return identical reward for all generated events), or mistakenly model the mappings between the generated events and the sentences in story as one-to-one. The specific discussions have been written in the line259~272 and Appendix A.5 of our paper. As a result, we propose a novel OT reward to effectively measure the quality of generated events. Also in the camera-ready version, we will supplement the details of IPOT and explain how to utilize IPOT to compute the optimal transport matrix (T).
>
> (5) Regarding the computation of story loss when the sketch is model-generated.
> ​	​	Res: Thank you for your detailed comments. The story loss is actually $L_{\theta}$ in the equation 3 of our paper. Specifically, let $S_i$ , $Y_i$ denote the $i$-th generated sketch and $i$-th golden story, respectively. The story loss is computed by summing up the negative log-likelihood loss of all included tokens as below:  $$L_{\theta} = -log(\hat{Y_i}|S_i)=-\sum_{k=1}^{|\hat{Y_i}|}logp(\hat{Y_{i,k}}|S_i,\hat{Y_{i,<k}})$$ We will provide a detailed description of the process of computing the story loss in the camera-ready version.
>
> (6) Regarding the equation 8 and equation 9 of our paper.
> ​	​	Res: Thank you for your detailed comments. Actually, there is no apparent connection between equation 8 and equation 9. Formally, they utilize the optimal transport matrix ($T$) to compute the reward in reinforcement learning and the optimal transport loss, respectively. The purpose of equation8 is to make reward consider the one-to-many mappings between the generated events and sentences in story. The specific details are illustrated in the upper left corner of Figure 3 and Appendix A.5. After executing equation 8, the reward for $i$-th event is not only determined by the $i$-th sentence but also by other sentences. This is because $r_i$ is the weighted sum of $L_s=\\{L_{s1}, L_{s2}, ..., L_{sk}\\}$ rather than only $-L_{si}$. As a result, one event will receive different rewards from different sentences in story so that the mapping is one-to-many (one event to many sentences). The purpose of equation9 is to measure the cost of moving the event semantic distribution in the sketch to sentence semantic distribution in the story. It is subsequently used to optimize the whole network, pushing these two distributions closer in semantics [1].
> [1] Generative Modeling with Optimal Transport Maps.  2022 ICLR

---

### Official Review · Reviewer_bja9 · 2023-08-06

**Soundness:** 3

**Excitement:**

3: Ambivalent: It has merits (e.g., it reports state-of-the-art results, the idea is nice), but there are key weaknesses (e.g., it describes incremental work), and it can significantly benefit from another round of revision. However, I won't object to accepting it if my co-reviewers champion it.

**Paper Topic And Main Contributions:**

The paper introduces the BPOT (Bidirectional Pretraining Model with Optimal Transport Reward) framework for story generation, aiming to tackle the challenges related to event correlations and narrative order in existing storytelling systems. To address these challenges, the framework utilizes a bidirectional pretrained model, which captures event correlations and pairwise event order. Additionally, a reinforcement learning algorithm with a novel optimal transport reward is employed to enhance the quality of the generated events. The results showcase the superior performance of the framework in generating logically coherent stories with flashback.

**Questions For The Authors:**

The overall experimental setup of the paper is relatively weak, with few baseline models. It is hoped that more comparative experiments, such as [1], can be added.

[1] Follow the Timeline! Generating an Abstractive and Extractive Timeline Summary in Chronological Order

**Reasons To Accept:**

The motivation and experiment setting is reasonable.

**Reasons To Reject:**

The overall experimental setup of the paper is relatively weak, with few baseline models. It is hoped that more comparative experiments, such as [1], can be added.

[1] Follow the Timeline! Generating an Abstractive and Extractive Timeline Summary in Chronological Order

**Reproducibility:**

3: Could reproduce the results with some difficulty. The settings of parameters are underspecified or subjectively determined; the training/evaluation data are not widely available.

**Reviewer Confidence:**

3: Pretty sure, but there's a chance I missed something. Although I have a good feel for this area in general, I did not carefully check the paper's details, e.g., the math, experimental design, or novelty.

---

> ### Author Rebuttal · Authors · 2023-08-28
>
> Thank you very much for your positive feedback.
>
> (1) Regarding the addition of a baseline.
>     ​	​    ​	Res: Thank you for your constructive suggestions. we carefully study this meaningful work [1], which proposed a unified abstractive and extractive timeline summarization framework. It first leverages a graph-based encoder to model the mutual influence among events. Then, it innovatively designs the time-event memory and evolutionary attention to learn temporal information and guide the generation. With that in mind, we integrate their core innovations into the "vanilla" version of our model to construct a corresponding baseline. The specific details are as follows:
> * First, we utilize a pretrained sketch model to generate sketches with temporal prompts. Later, we input the generated sketches into the BART encoder to extract word representation and event representation from the last hidden state of the encoder.
> * Next, following [1], we adopt a graph-based encoder to encode the global event representation.
> * Then, following [1], we denote the representation of temporal prompts as the key of the time-event memory as it explicitly expresses time series information. We also serve the local event representation and global event representation as the value of the time-event memory.
> * Last, following [1], we replace bart decoder with RNN and realize the evolutionary attention in the process of decoding.
>
> Due to time constraints, we have not yet completed the experiments. It is because RNN can't perform parallel operations like Transformer, resulting in lower efficiency. Actually, when serving Transformer as the decoder, it takes about 30 hours to complete the experiment on ROCStories as reported in our paper. However, it is estimated to take about 3.5 days when serving RNN as the decoder.  Furthermore, there are some essential differences between our work and [1]. Specifically, their temporal modeling is monotonic (always "from the past to the future"), while our temporal modeling is diverse (flashback includes "go back to the past"). Therefore, our temporal modeling is inherently more complex and challenging as it models more diverse temporal relations. Besides, we also consider event correlations to help temporal modeling. In the final version, we will complete the experiment and cite this meaningful work [1] as our additional baseline model.
> [1] Follow the Timeline! Generating an Abstractive and Extractive Timeline Summary in Chronological Order 2022 TOIS

---

### Official Review · Reviewer_qVqz · 2023-08-07

**Typos Grammar Style And Presentation Improvements:** Some typos or grammatical errors in L…
**Soundness:** 3

**Excitement:**

3: Ambivalent: It has merits (e.g., it reports state-of-the-art results, the idea is nice), but there are key weaknesses (e.g., it describes incremental work), and it can significantly benefit from another round of revision. However, I won't object to accepting it if my co-reviewers champion it.

**Paper Topic And Main Contributions:**

This paper investigates the claimed problem of narrative order aware story generation in a two-stage paradigm, i.e., pre-training the model with novel event-level mask pridiction tasks and fine-tuning the model with a OT reward based RL algorithm.

**Reasons To Accept:**

1. The overall work is solid and the experiments are sufficient.

2. The problem is interesting and so is the solution, especially the event masking tasks.

3. The paper writing is qualified to clearly introduce the work.

**Reasons To Reject:**

1. The authors claim that the proposed model can handle the narrative order and build the flashback stories. However, in my opinion the keys of the proposed method, BPOT, lie on the event masking based prediction tasks and the RL based fine-tuning, which both do not embody the motivation for building the flashback stories. Also the experiments cannot prove the flashback ability, instead of just showing the improvement of general story generation quality.

2. In terms of building the narrative order ability, I think it does not just mean the aspect of event. Other aspects include sentiment, role, storyline and so on. So the authors may over-claim the concept of ‘narrative’. I think this paper should be focused only within the scope of ‘event aware story generation’.

**Reproducibility:**

3: Could reproduce the results with some difficulty. The settings of parameters are underspecified or subjectively determined; the training/evaluation data are not widely available.

**Reviewer Confidence:**

4: Quite sure. I tried to check the important points carefully. It's unlikely, though conceivable, that I missed something that should affect my ratings.

---

> ### Author Rebuttal · Authors · 2023-08-28
>
> Thank you very much for your positive feedback and constructive comments.
>
> (1) Regarding the relation between our proposed method and building the flashback stories.
>     ​	​    ​	Res: Thank you for your detailed comments. In this paper, we adopt the temporal prompt (*<after>*) to reverse the narrative order to describe the past event for flashback generation (e.g., given "have a rest *<after>* <Mask>", the "<Mask>" might be infilled with flashback "feel tired" as "feel tired" occurs earlier than "have a rest"). However, building flashback stories requires generating high-quality events that balance the logic and the narrative order rather than simply adopting temporal prompts in the specified position. To better balance the logic and the narrative order, we propose a bidirectional pretraining model. It pretrains event blank infilling (order2event direction: aims to reason event from pairwise event order) and pairwise order prediction (event2order direction: aims to reason order from pairwise events) to facilitate the model to more sufficiently understand the event correlations and event temporal order. To generate higher-quality events, we propose a reinforcement learning algorithm where a novel optimal transport reward is designed to effectively measure the quality of generated events. It aims to guide the model to improve the quality of events by punishing low-quality events that receive low rewards. Through extensive experiments, it can be observed that our method can improve the flashback ability and outperform other methods (88.18% vs. 72.73% flashback accuracy in Figure 4 and 4.113 vs. 2.923 coherence score in Table 4). We believe that our method is highly correlated with generating flashback stories.
>
> (2) Regarding proving the improvement of flashback ability through experiment.
>     ​	​    ​	Res: Thank you for your detailed comments. In section 4.3 of our paper, we conduct a manual evaluation and specially design two metrics: Narrative Order Diversity (NOD, the higher the NOD, the more flashback stories are generated) and Narrative Order Accuracy (NOA, the higher the NOA, the higher the efficiency of generating flashback stories) to evaluate the flashback ability of different methods. The results in the first two columns of Table 4 demonstrate that our method consistently outperforms other baselines and ablation variants. Specifically, our method achieves the highest score on NOD (1.115) and NOA (0.958). To further evaluate the flashback ability of different methods, the statistical results of various narrative orders ("after" represents flashback) are presented in Figure 4. In the last bar of Figure 4, it can be observed that our method achieves the highest after ratio (24%) and highest after accuracy (88.18%), which confirms the strong flashback ability of our method. Additionally, a series of case studies are provided in Table 6 and Table 7, showcasing the generated flashback stories by our method. All of the above proves that our method can improve the flashback ability and outperform other methods.
>
> (3) Regarding building the narrative order and the overemphasis on the concept of "narrative".
>     ​	​    ​	Res: Thank you for your constructive suggestions. To the best of our knowledge, building the narrative order is a relatively new issue that has hardly been covered in previous research. It is also indeed related to many aspects (e.g., sentiment, role). In this paper, we follow the pioneering work [1], which reduces building the narrative order to modeling the pairwise event order. Besides, we also consider event correlations to help build the narrative order. The case studies in Table 6 and Table 7 indicate that the pairwise event order can effectively influence the narrative order, which proves the rationality of this simplified modeling approach. Additionally, in the final version, we will pay more attention to distinguishing the differences between narrative order and pairwise event order, avoiding overemphasizing "narrative". A discussion of this simplified modeling approach will also be added in the "Limitation" section. In the future, we will consider more aspects to comprehensively build the narrative order.
> [1] Go Back in Time: Generating Flashbacks in Stories with Event Temporal Prompts.  2022 NAACL
>
> (4) Regarding the reproducibility and typo error.
>     ​	​    ​	Res: Thank you for your detailed comments. Due to the double-blind policy, we can't directly post our code in the article, but we have submitted our code as an attachment to the review system. Additionally, we promise to make the code publicly available and describe more details for reproduction. We will also thoroughly proofread our paper and correct typos or grammatical errors in the final version.

---

### Meta-Review · Area_Chair_rZRm · 2023-09-20

**Recommendation:** 3

**Metareview:**

The work presents a novel method of keeping generating stories while keeping narrative order in mind through a two step pretraining and finetuning with RL+optimal transport rewards. All the reviewers agree that the method is sound and though the experimental setup is rather weak and can be improved. I would encourage the authors to update the paper with such results as they have stated in their rebuttal to reviewer bja9.

(Also a minor suggestion to edit the title Narrative Order Aware Story Generation via a Bidirectional Pretrained Model and Optimal Transport Reward)

---

### Decision · Program_Chairs · 2023-10-07

**Decision:**

Accept-Findings

**Comment:**

The work presents a novel method of keeping generating stories while keeping narrative order in mind through a two step pretraining and finetuning with RL+optimal transport rewards. All the reviewers agree that the method is sound and though the experimental setup is rather weak and can be improved. I would encourage the authors to update the paper with such results as they have stated in their rebuttal to reviewer bja9.

(Also a minor suggestion to edit the title Narrative Order Aware Story Generation via a Bidirectional Pretrained Model and Optimal Transport Reward)